# Role of Glucocorticoids and Glucocorticoid Receptors in Glaucoma Pathogenesis

**DOI:** 10.3390/cells12202452

**Published:** 2023-10-14

**Authors:** Pinkal D. Patel, Bindu Kodati, Abbot F. Clark

**Affiliations:** Department of Pharmacology & Neuroscience, North Texas Eye Research Institute, University of North Texas Health Science Center, Fort Worth, TX 76107, USA; pinkal.patel@unthsc.edu (P.D.P.); bindu.kodati@unthsc.edu (B.K.)

**Keywords:** glucocorticoid receptor, glaucoma, animal models, anti-inflammatory steroids, primary open-angle glaucoma, steroid glaucoma, ocular hypertension, eye, corticosteroids

## Abstract

The glucocorticoid receptor (GR), including both alternative spliced isoforms (GRα and GRβ), has been implicated in the development of primary open-angle glaucoma (POAG) and iatrogenic glucocorticoid-induced glaucoma (GIG). POAG is the most common form of glaucoma, which is the leading cause of irreversible vision loss and blindness in the world. Glucocorticoids (GCs) are commonly used therapeutically for ocular and numerous other diseases/conditions. One serious side effect of prolonged GC therapy is the development of iatrogenic secondary ocular hypertension (OHT) and OAG (i.e., GC-induced glaucoma (GIG)) that clinically and pathologically mimics POAG. GC-induced OHT is caused by pathogenic damage to the trabecular meshwork (TM), a tissue involved in regulating aqueous humor outflow and intraocular pressure. TM cells derived from POAG eyes (GTM cells) have a lower expression of GRβ, a dominant negative regulator of GC activity, compared to TM cells from age-matched control eyes. Therefore, GTM cells have a greater pathogenic response to GCs. Almost all POAG patients develop GC-OHT when treated with GCs, in contrast to a GC responder rate of 40% in the normal population. An increased expression of GRβ can block GC-induced pathogenic changes in TM cells and reverse GC-OHT in mice. The endogenous expression of GRβ in the TM may relate to differences in the development of GC-OHT in the normal population. A number of studies have suggested increased levels of endogenous cortisol in POAG patients as well as differences in cortisol metabolism, suggesting that GCs may be involved in the development of POAG. Additional studies are warranted to better understand the molecular mechanisms involved in POAG and GIG in order to develop new disease-modifying therapies to better treat these two sight threatening forms of glaucoma. The purpose of this timely review is to highlight the pathological and clinical features of GC-OHT and GIG, mechanisms responsible for GC responsiveness, potential therapeutic options, as well as to compare the similar features of GIG with POAG.

## 1. Introduction

Glaucoma, the leading cause of irreversible blindness worldwide, is a heterogenous group of optic neuropathies characterized by neuronal degeneration in the retina and the brain. In 2020, nearly 80 million people had glaucoma worldwide, and this number is expected to increase to over 112 million by 2040, imposing a major burden on global health care [1,2]. The debilitating vision loss in glaucoma is primarily due to optic nerve axon degeneration and the progressive loss of retinal ganglion cells, which are neurons that integrate visual signals from the photoreceptors and generate action potentials that are relayed to visual centers in the brain. There are several risk factors associated with the development of glaucoma, including age (>40 years), elevated intraocular pressure (IOP), ethnicity, family history, myopia, and glucocorticoid (GC) responsiveness. IOP is a major causative risk factor and the only treatable one for both the development [3] and progression [4] of glaucoma. African Americans are 6–8 times more likely and Hispanics 2–3 more likely to develop glaucoma compared to Caucasians [5,6]. First-degree relatives of individuals with glaucoma are at higher risk of also developing glaucoma [7,8]. Myopia associated with the elongation of the eye globe is a glaucoma risk factor, perhaps due to the altered geometry of the optic nerve head (ONH). Prolonged therapies with GCs can cause ocular hypertension in some individuals, and these steroid responders are at a much higher risk of developing glaucoma [9,10] and vice versa [11]. Also, the prevalence of glaucoma is greater in urban than in the rural population [12], perhaps due to the air pollution associated with large cities [13].

Glaucoma is categorized into subsets including: primary open-angle glaucoma (POAG), primary closed-angle glaucoma (PACG), secondary open-angle glaucoma, and congenital glaucoma. Both POAG and PACG occur due to the mechanical obstruction in the trabecular meshwork (TM), which plays a major role in the drainage of the aqueous humor out of the anterior segment, thereby regulating IOP. Developmental abnormalities in the anterior chamber of the eye result in congenital glaucoma and juvenile glaucoma. The use of certain medications such as GCs, exfoliation syndrome, pigment dispersion, and eye injury can cause secondary open-angle glaucoma. Glaucoma is slowly progressive, usually painless, and asymptomatic in the initial stages. However, as the disease progress, patients begin to lose the peripheral vision that can progress to central vision loss and blindness. The most common type of glaucoma is POAG. The prevalence of POAG is highest in Africa, Europe, and the Americas, while the prevalence of PACG is highest in Asia.

Although lowering IOP by pharmaceutical or surgical therapies often is effective in slowing the disease progression, the resulting neurodegeneration is irreversible. Along with the mechanical stress on the ONH due to IOP elevation, retinal vascular dysregulation/insufficiency also plays a role in damaging the ONH and retinal ganglion cells (RGCs) [14]. Vascular dysregulation may occur by defective autoregulation and diminished blood flow, which compromises the metabolic needs of RGCs and the ONH [15,16]. In patients with POAG and animal models of glaucoma, endothelin-1 levels are elevated in the aqueous humor as well as the circulation. Endothelin has strong vasoconstrictive activities that would decrease ocular perfusion and generate ischemic conditions [17].

POAG involves the dysfunction of several ocular tissues (Figure 1). Biomechanical stress and altered cytokine signaling in the TM results in abnormalities of the normal aqueous humor outflow pathway. In glaucoma, there is an increase in the thickness of the elastic fiber network and anterior elastic tendons of the ciliary muscle along with the extracellular matrix (ECM) deposition in the TM. Due to the loss of TM cells, there is a fusion and thickening of the trabecular lamellae, a narrowing of intertrabecular spaces and collector channels, and a thinning of Schlemm’s canal lumen [18,19]. The actin cytoskeleton in the TM and ONH becomes reorganized, which further contributes to the pathophysiology of glaucoma [20,21]. It should be noted that these same glaucomatous changes to the TM also occur in GC-OHT (see Section 3 for details). Axonal damage at the ONH causes the progressive loss of RGCs and optic nerve axons with characteristic defects in the visual field. The visual pathway consists of two parts, the anterior visual pathway (the retina, optic nerve, and optic chiasm and lateral geniculate nucleus (LGN)) and the posterior visual pathway (optic radiations and visual cortex) [17]. The primary site of damage in glaucoma is at the optic nerve head region, consisting of RGC axons, blood vessels, connective tissues, and glia. At this region, the perforated sclera is more susceptible to the damage, which results in the lamina cribrosa deformation and remodeling, and these changes progressively decrease axonal transport [22]. The deprivation of brain-derived neurotrophic factor may result from retrograde axonal transportation deficits. The high energy demands for the proper functioning of these neurons cannot be maintained, and ultimately lead to the neurodegeneration of RGCs and their axons. The clinical manifestation of glaucoma includes an increased optic cup-to-disk ratio, reduced retinal nerve fiber layer thickness, notch at the neural rim, hemorrhage at the optic disc margin, and characteristic visual field defects [23].

## 2. Glucocorticoids and the Eye

The normal physiological activities of glucocorticoids (GCs) are to regulate carbohydrate, lipid, protein, and electrolyte metabolism. The endogenous GC in man is cortisol, which is secreted by the adrenal cortex in response to physiological stresses. Cortisol also has anti-inflammatory and immunosuppressive activities, which has led to the design, synthesis, and development of more potent and effective GCs with longer half-lives to treat a wide variety of inflammatory and immune diseases. GCs are unsurpassed as anti-inflammatory agents because they inhibit most stages of the inflammatory response, irrespective of the cause of inflammation. There exists a complex relationship between ocular inflammation and IOP. The IOP depends on the comparative rates of aqueous production and outflow. Ocular inflammation can negatively affect these processes and upend normal circulation in the eye, leading to IOP elevation. In inflammatory forms of glaucoma, GCs can resolve IOP elevation by reducing trabecular inflammation that improves TM outflow and increasing blood–aqueous barrier (BAB) to decrease aqueous viscosity. On the other hand, GCs can increase IOP by reducing ciliary body inflammation and improving aqueous secretion. In susceptible individuals, prolonged GC treatment can also cause increased resistance to aqueous outflow by inducing pathological changes in TM cells, which leads to IOP elevation called ocular hypertension (OHT).

## 3. Glucocorticoid-Induced Ocular Hypertension and Glucocorticoid-Induced Glaucoma

Ocular hypertension (OHT) or elevated intraocular pressure (IOP) can occur after GC use in susceptible individuals of any age group. A variety of ocular conditions are treated by GC therapy (Table 1). The prevalence of GC-OHT in patients treated therapeutically with GCs can vary widely depending on the specific GC, its potency, route of administration, penetration into the anterior segment, and duration of treatment. However, the delivery of GCs into the vitreous cavity either via injection of a suspension or delivery of a slow-release implant leads to greater responder rates of 30% (ivt injection of TA suspension) up to 70% (with slow-release implants). In some cases, patients are recalcitrant to topical glaucoma therapy and proceed to glaucoma surgery [24].

Long-term GC therapy in these susceptible individuals can lead to prolonged IOP elevation causing glucocorticoid-induced glaucoma (GIG), which is a secondary form of iatrogenic open-angle glaucoma. Apart from GIG, the IOP elevation can be observed in differential diagnoses of several types of primary and secondary glaucomas, which include POAG, angle-closure glaucoma, angle-recession glaucoma, pigmentary glaucoma, plateau iris glaucoma, pseudoexfoliation glaucoma, and uveitic glaucoma [25]. However, one characteristic differentiating GIG from other forms of glaucomas is that IOP elevation by GC is generally reversible upon discontinuation of GC therapy.

GC-induced iatrogenic secondary open-angle glaucoma clinically and molecularly mimics POAG (Table 2 and Figure 2). In fact, differential diagnosis often includes the evaluation of prior or current GC therapy. Ocular hypertension in both GC-OHT and POAG are due to increased aqueous humor outflow resistance, and both are associated with similar structural and molecular changes to the trabecular meshwork (TM) (Figure 3 and Table 2). These glaucomatous changes to the TM include: the deposition of ECM material within the trabecular beams and within the JCT (due to increased ECM synthesis, increased crosslinking, and decreased degradation), the reorganization of the actin cytoskeleton (CLANs), progressive loss of TM cells, and increased tissue stiffness. This ocular hypertension progressively damages unmyelinated RGC axons at the ONH, the fibrosis of the ONH, blocking anterograde and retrograde axonal transport, the degeneration of the optic nerve, and the death of RGCs and target neurons in the vision centers of the brain.

## 4. Glucocorticoid Receptors (GRs)

The physiological and pharmacological effects of GCs are mediated by the GC receptor (GR), which is an intracellular, ligand-dependent, nuclear transcription factor. The GR protein is composed of structural and functional domains. These domains include: a transactivation domain on the amino-terminus responsible for gene activation; a DNA-binding domain in center to bind specific regions on the DNA called glucocorticoid response element (GRE); and a ligand-binding domain (LBD) on the carboxy terminus that enables binding to GCs. The carboxy terminus of GR also contains regions involved in nuclear translocation, receptor dimerization, protein binding, and transactivation. In TM cells, GR affects the expression of hundreds of genes. The full-length GR is alternatively spliced in exon 9 to generate two isoforms of the GR differing in their carboxy terminus (Figure 4). GRα binds GCs with high affinity and is the major biological and pharmacological receptor mediating the transcriptional effects of GCs. In contrast, GRβ does not bind GCs and acts as a dominant negative regulator of GC activity. Further heterogeneity in these GRs is mediated by alternative N-terminal start sites leading to differences in the N-terminal domain length [27].

GCs work by entering cells and binding the cytosolic glucocorticoid receptor (GRα) complex. Upon binding the GC, GRα undergoes a conformational change, which dissociates the activated GRα from the accessory proteins. The activated GRα is translocated into the nucleus, where it homodimerizes and binds to GREs to directly alter gene expression (transactivation). The activated GRα can also bind to other transcription factors (e.g., AP-1, NFκB) thereby preventing these other transcription factors from binding to their response elements and suppressing transcription (transrepression). Therefore, GRα acts as a ligand activated transcription factor, and most of the biological activities of GCs are mediated through altered gene expression (Figure 4). Almost all cells in the body contain GRs, but each cell type responds differently to GCs based on the chromatin structure and cell-specific transcription factors.

## 5. Steroid Responders: Prevalence and Associated Risk Factors

Individuals susceptible to IOP elevation with GC treatment are known as steroid responders (SR). Steroid responsiveness has been defined differently in many studies, but the clinical consensus dictates that an IOP of 21–24 mmHg or an increase over baseline IOP of >5–10 mmHg is to be considered a steroid response [28,29,30]. Although the exact prevalence of SR in the population varies depending on the specific GC, the route of administration, and duration of therapy, approximately 30–40% of the population are steroid responders. Early studies categorized the level of steroid responsiveness into three main categories: non-responders, moderate responders, and high responders. A majority 2/3 of the population is considered non-responder with IOPs after GC treatment less than 20 mmHg or an IOP rise of <6 mmHg over baseline. Moderate responders comprise a third of the population with an IOP elevation of 25–31 mmHg or an increase of 6–15 mmHg over baseline. Finally, approximately 4–6% of the population are high responders with GC-mediated rise in IOP above 31 mmHg or more than 15 mmHg elevation over baseline. There are several risk factors associated with the development of steroid responsiveness; chief among these is the diagnosis or family history of primary open-angle glaucoma (POAG) with an estimated 90% of POAG patients being high GC responders. Other risk factors include elevated IOP, diabetes mellitus, myopia, connective tissue disorders (rheumatoid arthritis), long-term use of steroids, and age. Steroid response follows a bimodal distribution in the context of age, where older adults and children are at higher risk of developing GC-OHT [31,32].

## 6. Properties of GCs That Lead to OHT

The timeline and the severity of IOP elevation depends on multiple factors including the anti-inflammatory potency, dose, duration of treatment, and route of GC administration. The effect and potency of GC significantly depends on the ring structure or the side groups of the steroid base molecule. Chemists have discovered and developed newer and more potent glucocorticoids for the treatment of inflammatory diseases. The overall goal in designing these new GCs has been to remove the mineralocorticoid activity of cortisol, improve the anti-inflammatory activity, and increase the potency. The potency of GCs is a major factor in determining the likelihood of developing GC-OHT. For example, dexamethasone and prednisolone are more potent steroids and increase IOP more frequently than less potent steroids such as hydrocortisone (cortisol). There are several different types of GCs with varying levels of potency used in treating ocular diseases [33]. Table 3 lists common GCs used for ocular therapy, their concentrations, and routes of delivery. Table 4 lists topical ocular GCs, their relative potency compared to cortisol, and the magnitude of ocular hypertension induced by each. These synthetic compounds alter the bioavailability and potency, thereby affecting penetration into the eye, release characteristics, and metabolism.

The route of GC administration is another important aspect to consider in GC therapy. GCs can be administered to the eye in multiple different ways (Figure 5) depending on the disease and treatment requirements. The routes of GC delivery include topical, periocular, intravitreal or intracameral injections, and systemic administration. Topical formulations, applied in the form of eyedrops or ointments, penetrate the anterior segment but have poor penetration of the posterior segment. Therefore, topically administered GC formulations are used for inflammatory diseases of the ocular surface and anterior segment. The periocular route of delivery involves subconjunctival, sub-Tenon’s, inferior trans-septal, or retrobulbar injections. These routes form depots of longer acting GCs in the periocular space, which penetrate the eye over several weeks. Patients administered via periocular route are more likely to develop GC-OHT than those treated topically. GCs can be also delivered intravitreally by the injection of a suspension (triamcinolone acetonide), a degradable implant (dexamethasone intravitreal implant), or by the surgical implantation of a sutured sustained release device (fluocinolone acetonide). Intravitreal injections tend to have a short duration of action (months) compared to a sustained-release implant that releases the GC over a period of 2 ½ years. The systemic delivery of GCs has been indicated for treatment of posterior segment ocular diseases. Due to the normal blood–retinal barrier (BRB) and blood–aqueous barrier (BAB), the ocular bioavailability of systemically delivered GC is low, and as a result, a higher dose is typically administered. However, there is a breakdown of the BAB or the BRB in some diseases (e.g., uveitis and macular edema), which enhances the GC delivery to the diseased tissues. As previously mentioned, increasing the dose of GC may lead to serious systemic side effects as well as ocular side effects that include the development of cataracts and elevated IOP.

## 7. Mechanism of GC-Induced IOP Elevation

The clinical and pathological manifestations of GIG closely resemble POAG (Table 2 and Figure 2). Elevated IOP and glaucomatous damage at the optic nerve head (ONH) is observed in both diseases. Like POAG, IOP elevation in GIG is asymptomatic and often incidentally detected by ophthalmologists during a routine clinical examination. In severe cases, this disease can remain undetected until the patient experiences blurred vision or visual field loss. A patient′s medical history provides a valuable clue to the underlying etiology and determination of proper diagnosis. Individuals treated with GCs should have their IOP routinely monitored.

GCs are unsurpassed in their anti-inflammatory and immunosuppressive activities because they intervene in inflammatory and immune responses at multiple levels. This makes GCs one of the most commonly prescribed therapies for diseases and conditions associated with inflammation. Likewise, often the side effects of GC therapies are due to multiple mechanisms. Despite considerable research, the exact molecular mechanism(s) responsible for GC-induced IOP elevation observed in GIG remains unknown. However, the IOP elevation in GIG is primarily due to pathological damage to the conventional outflow pathway including molecular and morphological changes to the TM. TM cells express both GC receptor isoforms, which are essential for steroid responsiveness. GCs elicit a wide variety of effects on TM cells and tissues, which together damage the outflow pathway and thereby elevate IOP in GIG [26,36].

The TM functions as a biological sieve, which filters and drains the aqueous humor out of the eye while maintaining optimal physiological IOP. GCs induce a number of changes to TM cells (Figure 3). In TM cells and ex vivo cultured human TM tissues, GCs induce fibrotic changes including the increased deposition of ECM in the extracellular space, which in turn, increases resistance aqueous humor flow through the meshwork. This results in reduced aqueous humor outflow and elevation of IOP. GCs contribute to increased TM resistance by increasing expression and extracellular deposition of fibronectin, glycosaminoglycans (GAGs), elastin, collagen, and laminin within the trabecular meshwork tissue [26,36]. Furthermore, GCs have been shown to stiffen TM cells and tissues in multiple species by reorganizing the actin cytoskeleton, which includes the increased formation of F-actin stress fibers, an increased expression of α-smooth muscle actin in addition to increased ECM deposition. The GC treatment of TM cells and tissues also leads to the formation of cross-linked actin networks (CLANs) [37,38], which are geodesic dome-like actin structures characteristic to glaucomatous TM cells [20] and tissues [39]. This cytoskeletal reorganization and resulting stiffening of the TM tissues can have a detrimental consequence on the TMs’ ability to phagocytose and filter cellular debris from the aqueous humor, which can further exacerbate the GIG pathology [40].

## 8. Role of Endogenous Cortisol and Cortisol Metabolism in POAG

Several studies have suggested that the endogenous levels of the natural glucocorticoid cortisol cause ocular hypertension associated with POAG. This proposal is based on the findings of elevated plasma [41,42,43,44,45] and aqueous humor [41,45] cortisol in POAG patients. There also are reports of altered cortisol responses in POAG patients [46,47,48,49,50]. In addition, Southren and colleagues reported aberrant cortisol metabolism in POAG patients [51,52]. The hypothesis is further supported by the findings that individuals who are steroid-responders have a higher risk of developing POAG [9,10]. However, other labs were not able to replicate elevated plasma cortisol or altered cortisol sensitivity in POAG patients [53,54], which made the association between endogenous cortisol and elevated IOP less clear. It should be noted that serum levels of cortisol vary widely depending on the time of day as well as the varying levels of stress in individuals, which may well explain the different findings among these studies.

## 9. Models of Glucocorticoid-Induced Ocular Hypertension

GC-OHT is not specific to humans, and a variety of models and species have been used to better understand this important ocular side effect of GC therapies (Table 5). The question of whether GCs directly or indirectly cause OHT was resolved using an ex vivo model of perfusion cultured human anterior segments [38,55]. Paired anterior segments from human donor eyes were mounted on specially engineered dishes that clamp the anterior segments to make a watertight seal. Chemically defined culture medium was perfused into the anterior chamber at the rate of normal aqueous humor formation (2.5 μL/min) and exits the eye through the TM and Schlemm’s canal. A second canula is connected to a pressure transducer to continuously monitor IOP. Dexamethasone (100 nM in 0.1% ethanol) was added to the perfusion medium of one eye while the contralateral eye received vehicle (0.1% ethanol), and the paired eyes were perfusion cultured for 14 days. This dose of DEX was selected based on human clinical PK studies demonstrating an aqueous humor concentration of approximately 100 nM following a single ocular drop of a 0.1% DEX suspension [56]. Thirty percent of the DEX-treated eyes in our ex vivo model developed a significantly elevated IOP (≥5 mmHg increase from baseline). Interestingly, the responder rate in this ex vivo model matched that seen clinically in man. This model has been used to evaluate the morphological, biochemical, and molecular changes associated with GC-OHT [38,55].

Nonhuman primates: We have shown that non-human primates also develop DEX-induced OHT [57]. Cynomolgus monkeys were topically dosed with a clinical formulation of 0.1% DEX three times per day for 3 weeks. Five of the eleven animals developed a statistically significant IOP elevation (≥5 mmHg above baseline) equating to a responder rate of 45%, which is close to that seen clinically in man. As in man, IOP returned to baseline 7 days following the withdrawal of DEX treatment. This experiment was repeated, and the animals again received the same topical ocular dosing regimen, and the same responders again developed DEX-OHT, while the previous non-responders continued to be non-responsive and did not develop DEX-OHT.

Cows: TM cells isolated from bovine eyes are often used experimentally to better understand TM cell biology, including the effects of GCs on TM cell extracellular matrix [58,59], cytoskeleton [60,61], and gene expression [62,63]. Danias and colleagues evaluated the effect of the topical ocular administration of prednisolone (0.5% suspension) on IOP in cattle [64] and found that all animals developed significant ocular hypertension (100% responder rate). However, an independent study using ex vivo perfusion culturing bovine anterior segments showed that only 12 out of 29 eyes perfused with 100 nM DEX developed a significant OHT (40% responder rate) [65]. These differences in responder rates could be due to different experimental designs (i.e., in vivo vs. ex vivo; different GCs, etc.) as well as sources of subjects (i.e., in vivo study was performed in Argentina using Argentine cattle, while the ex vivo study was study on young Texas cattle (<6 months)).

Sheep: In addition to cows, sheep also have been tested for GC-OHT using topical ocular dosing with a clinical formulation of 0.5% prednisolone acetate three times per day for 3–4 weeks, with all sheep developing significantly elevated IOP (100% responder rate) [66]. This sheep GC-OHT model has been used to evaluate several different antihypertensive agents including anecortave acetate [67], tissue plasminogen activator (tPA) [68], and gene therapy with a GRE-MMP1 vector [69].

Cats and dogs: Cats and dogs also developed GC-OHT, which was initially seen when veterinarians treated these pets with ophthalmic GCs for ocular inflammation [70,71,72,73]. Several groups developed feline models of GC-OHT to better understand this clinically relevant condition. Zhan and colleagues treated adult mixed breed cats with topical ocular administration of 0.5% DEX, 1% DEX or 1% prednisolone acetate (each 2–3 times/day) for up to 80 days, and all three formulations induced significant OHT [74]. As seen clinically in man, IOPs returned to baseline within 7 days of discontinuing GC treatment. There were apparent differences in the onset and degree of OHT in cats treated with 0.5% DEX, with 8 out of 12 cats having greater DEX responses. The investigators also showed that topical ocular treatment with PGF2α-isopropyl ester significantly lowered IOP in these GC-OHT cats. Bhattacherjee and colleagues examined the effects of the topical ocular administration of five different ophthalmic GCs (0.1% DEX, 1% prednisolone acetate, 1% loteprednol etabonate, 0.25% fluorometholone, and 1% rimexolone) for > 30 days on IOPs in adult female cats [72]. Each GC varied in its ability to induce OHT with DEX > pred-Ac > loteprednol etabonate > fluorometholone > rimexolone. Elevated IOPs returned to baseline levels 3–7 days after discontinuing GC administration. This closely matched the propensities of these GCs to induce OHT in man. This model was also used to show that topical ocular treatment (3 times/day) with a DEX derivative (0.1% dexamethasone beloxil) having a lower propensity to raise IOP compared to 0.1% DEX [75].

Rabbits: Rabbits were the first species evaluated for GC-OHT after the initial discovery of this side effect in man, and this model was explored in multiple labs. Lorenzetti was one of the first to demonstrate that the topical ocular delivery of corticosteroids induced OHT in rabbits [76], and this discovery has been repeated in other labs around the world. A number of different glucocorticoids (i.e., DEX, betamethasone, cortisone, triamcinolone, fluoromethalone, rimexolone, loteprednol etabonate) administered by topical ocular, subconjunctival, and intravitreal injections elevate IOP in rabbits. Almost all of these studies have been conducted in New Zealand albino (white) rabbits; however, other rabbit strains including New Zealand red [77], New Zealand silver, red Bourgogne, black mongrel [78], Japanese white [79], and Dutch belted rabbits [80] also developed GC-OHT. IOP elevations (ΔIOP) have been reported from slightly more than 1 mmHg [81] to 10 mmHg [82], although the majority of studies show IOP elevations of 4–6 mmHg. Also, responder rates vary from 50 to nearly 100%. It appears that younger rabbits are more susceptible to GC-OHT [77,83].

Rats: The evaluation of GC-OHT in rodents advanced with the discovery of methods to non-invasively and accurately measure IOPs in rats [84] and mice [85]. One group has shown that the topical ocular administration of 0.1% DEX for 3–4 weeks in rats significantly elevated IOP [86,87,88]. A second group evaluated the IOP-lowering effects of the topical ocular administration of 0.2% trans-resveratrol in a similar rat model of DEX-OHT [89]. Razali and colleagues treated rats with the topical ocular administration of 0.1% DEX and reported a significant IOP elevation in 8/10 rats and showed damage to the TM as well as glaucomatous degeneration of RGCs and the thinning of the GCL and inner retina (i.e., GC-induced glaucoma) [89,90]. Different modes of GC administration have also been successful in the generation of GC-OHT. Subconjunctival injection of the potent GC betamethasone phosphate (30 μL containing 300 μg) also significantly elevated IOPs in young SD rats [91]. Anterior chamber injections of DEX-loaded PLGA microspheres at 0 and 4 weeks not only caused significant OHT but also the thinning of the RNFL and loss of RGCs [92]. However, a number of labs have been unsuccessful in generating GC-OHT in rats (personal communications), with one group reporting a considerable loss in body weight and IOP lowering with topical ocular administration of 0.1% DEX [93].

Mice: Whitlock and colleagues demonstrated that mice implanted with osmotic minipumps continuously delivering DEX systemically caused a significantly elevated IOP of approximately 3 mmHg [94]. They tested hybrid mice on a mixed background (B6.129) and found heterogeneity in DEX-OHT responsiveness. Overby and colleagues [95] also delivered DEX in osmotic minipumps to C57BL/6 mice and showed a 2–3 mmHg elevated IOP due to increased aqueous humor outflow resistance as well as ultrastructural changes to the TM and the outflow pathway of these mice. Unfortunately, systemic exposure to DEX caused significant losses in body weight despite being given high calorie chow, resulting in a 40% “dropout” rate after 3–4 weeks of DEX treatment [87]. Zode and colleagues dosed C57BL/6J mice with topical ocular 0.1% DEX three times a day for up to 6 weeks and showed significant IOP elevation beginning at 2 weeks with a larger ΔIOP of 6–7 mmHg at 6 weeks [96]. They also showed that this DEX-OHT caused optic nerve axon degeneration and the progressive structural and functional loss of retinal ganglion cells after 10-20 weeks of DEX exposure. The local topical delivery of DEX did not cause a significant loss in body weight compared to systemic DEX exposure. Recently, a less “labor-intensive” method has been developed to generate mouse GC-OHT. C57BL/6J mice received weekly bilateral periocular (lower fornix) injections of DEX-acetate, which provides a slow-release depot of DEX to significantly elevate IOP by 5 mmHg when measured at daytime, and 10–12 mmHg when measured at night [97,98,99]. This elevated IOP was correlated with statistically significant reductions in the aqueous outflow facility measured in live anesthetized mice [97,99]. This GC-OHT is due to transactivation as opposed to transrepression since the GC-OHT was not seen in GR^dim^ mice [98]. Mice treated with weekly periocular injections of DEX-acetate for 10 weeks developed glaucomatous optic neuropathy (i.e., optic nerve degeneration and transport defects, RGC structural and functional loss, and immune cell infiltration at the ONH) [100]. Wang and colleagues used this route of administration to demonstrate increased the TM stiffness associated with DEX-OHT [101].

## 10. Role of GRβ in Regulating DEX-OHT

The alternatively spliced GRβ isoform acts as a dominant negative regulator of GC activity, but is GRβ involved in regulating GC-responsiveness in the TM thereby regulating IOP? Human TM cells isolated from POAG and control donor eyes differ in their expression of GRβ. Normal TM cells (NTM) express both GRα and GRβ isoforms, while GRβ expression is negligible in glaucoma TM (GTM) cells [102]. GTM cells are “hypersensitive” to GCs compared to NTM cells, and GTM cells become less sensitive to GCs when transfected with a GRβ expression vector [40,102]. The alternative splicing of the GR in TM cells involves spliceosome proteins SFRS3, SFRS5, and SFRS9, and a lower expression of SFRS5.1 in GTM cells favors the expression of GRα over GRβ, thereby making GTM cells more sensitive to GCs [103].

We used our mouse model of weekly periocular DEX-acetate administration to determine the effects of GRβ on DEX-OHT [97]. Upon reaching the peak OHT (ΔIOP of 10–12 mmHg), mice were divided into three groups, all of which continued to receive weekly DEX-acetate injections. In the groups that were not transduced or transduced with control virus (Ad5.null), the elevated IOPs remained unchanged. However, in the group where the TM was transduced with Ad5.GRβ, IOPs returned to untreated baselines over the course of 7 days and continued at those lower IOPs, even though these mice continued to be treated with DEX-acetate. This suggests that gene therapy (i.e., select delivery of GRβ to the TM) could become a therapeutic option to treat those patients (e.g., with persistent and vision-threatening uveitis) who develop GC-OHT and secondary glaucoma.

## 11. Conclusions and Future Directions

GCs are unsurpassed in their anti-inflammatory and immunosuppressive activities and are very commonly prescribed for a wide variety of diseases and conditions, including numerous eye diseases. However, prolonged GC treatment can induce ocular hypertension (OHT) in susceptible individuals that can progress to an iatrogenic secondary open-angle glaucoma (GIG). Interestingly, GC-OHT leads to GIG clinically and molecularly mimicking POAG, and there have been reports suggesting a role for the endogenous GC cortisol in the pathogenesis of POAG. Elevated IOP in POAG and GC-OHT is due to pathogenic damage to the trabecular meshwork (TM), and a tissue at the iridocorneal angle that regulates IOP. The TM expresses both isoforms of the glucocorticoid receptor, GRα and GRβ, and the ratio of these two isoforms makes TM cells more responsive (low GRβ) or less responsive (high GRβ) to GCs. TM cells isolated from glaucoma donor eyes contain low levels of GRβ, making them highly responsive to GCs; similarly, almost all POAG patients develop GC-OHT upon GC therapy. There is a spectrum of GC responsiveness in the general population, and the molecular mechanisms for this different responsiveness in developing GC-OHT is currently unknown but may be due to differences in the expressions of the two GR isoforms. GC-OHT is not unique to man, and numerous other species also develop GC-OHT when administered GCs. These animal models of GC-OHT are allowing researchers to discover the mechanisms responsible for GC-OHT as well as develop novel disease-modifying therapies. For example, the transduction of the TM with a GRβ expression vector totally reverses GC-OHT in mice suggesting that gene therapy could be used to treat patients undergoing long-term GC therapy. Insights into the molecular mechanisms responsible for GC-OHT may also lead to the discovery of pathogenic pathways playing a role in the development of POAG.

## Figures and Tables

**Figure 1 cells-12-02452-f001:**
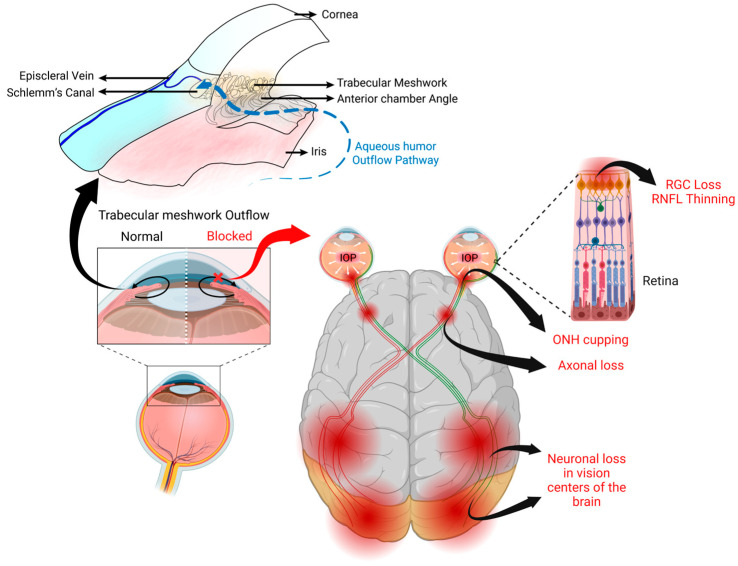
Sites in the eye involved in glaucoma pathogenesis. The major risk factor for the development and progression of glaucoma is elevated IOP due to impaired aqueous humor outflow at the trabecular meshwork, a tissue located at the iridocorneal angle. This elevated IOP causes the cupping of the optic nerve head (ONH), which damages the unmyelinated retinal ganglion cell (RGC) axons. This leads to anterograde degeneration, thinning of the retinal nerve fiber layer (RNFL), and death of the RGCs as well as the retrograde neurodegeneration of the optic nerve and target neurons in the vision centers of the brain. Figure created using BioRender.

**Figure 2 cells-12-02452-f002:**
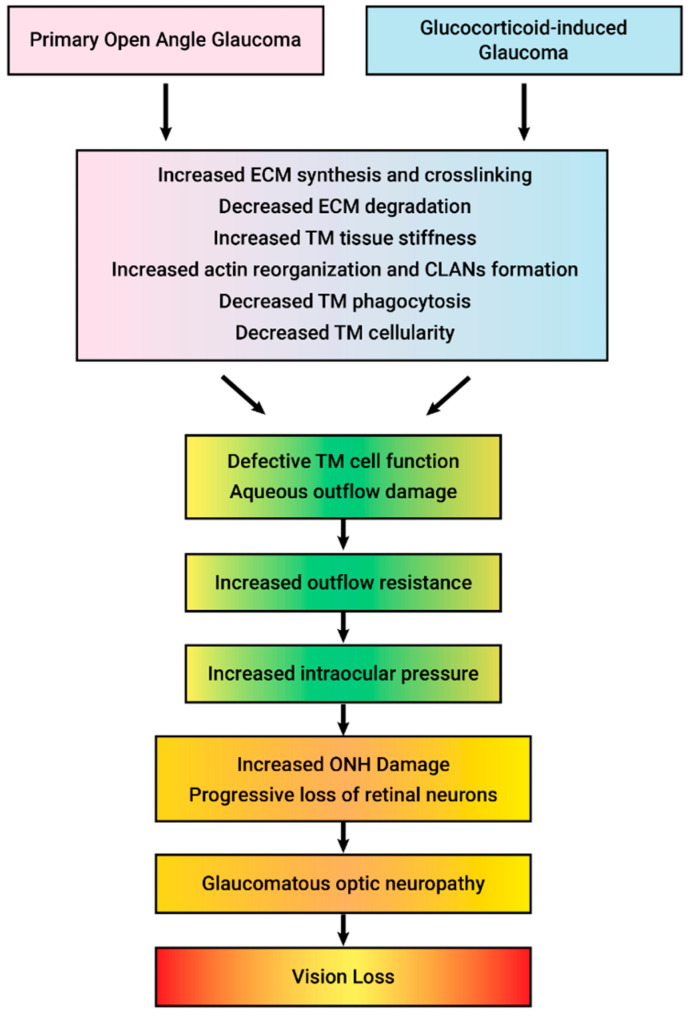
Clinical and molecular similarities between primary open-angle glaucoma (POAG) and glucocorticoid-induced glaucoma. In both diseases, the fibrosis of the trabecular meshwork (TM) increases aqueous humor outflow resistance, resulting in increased intraocular pressure that damages the optic nerve head causing optic neuropathy and loss of vision.

**Figure 3 cells-12-02452-f003:**
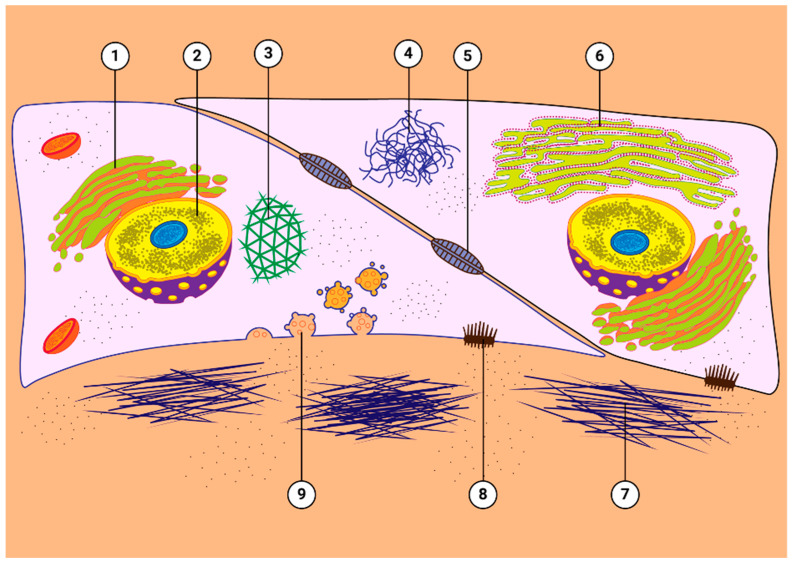
Effects of GCs on TM cells. GCs have a variety of effects on TM cells, including: (1) the proliferation of the Golgi apparatus; (2) enlarged and pleomorphic nuclei; (3) increased cross-linked actin networks (CLANs); (4) microtubule tangles; (5) altered expression and localization of cell–cell junctional complexes; (6) proliferation of rough endoplasmic reticulum; (7) increased deposition of extracellular matrix material around cells; (8) altered expression of integrins; and (9) increased numbers of electron lucent fusion vesicles. Figure derived from Wordinger and Clark [26].

**Figure 4 cells-12-02452-f004:**
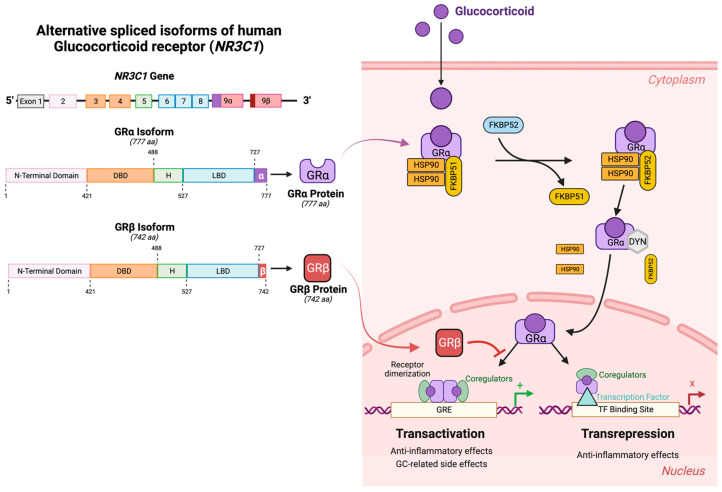
Alternative splicing of human GR and mechanisms of action of GRα and GRβ. (**Left**) The human GR gene (*NR3C1*) encodes 9 exons. The alternative splicing of exon 9 leads to isoforms: GRα (the active ligand binding transcription factor) and GRβ (dominant negative regulator of GC activities). DBD = DNA biding domain; H = hinge region; LBD = ligand-binding domain. (**Right**) Mechanisms of action of the GR isoforms. Unliganded GRα resides in the cytoplasm complexed with HSP90 and FKBP51. Upon binding GR, the accessory proteins dissociate, and activated GRα translocates to the nucleus. Once inside the nucleus, GRα homodimers recognize the glucocorticoid response elements (GREs) on the promoter regions of the GC responsive genes to directly alter transcription (transactivation). Alternatively, the activated GRα can bind to other transcription factors (e.g., AP-1, NFκB) to inhibit their binding of the corresponding response elements, thereby inhibiting transcription (transrepression). GRβ resides in the nucleus, blocking GRα-mediated transcriptional activities. Figure created using BioRender.

**Figure 5 cells-12-02452-f005:**
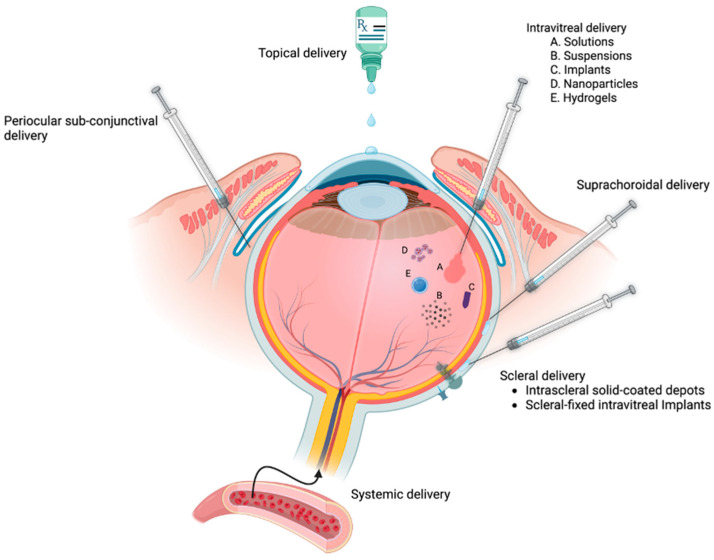
Various ocular delivery routes for anti-inflammatory GCs. Topical ocular delivery is used to treat the ocular surface and anterior segment. Drug depots can be placed adjacent to the eye by periocular sub-conjunctival injections, superchoroidal injections or intrascleral injections. Intraocular delivery is often used to treat posterior segment tissues (e.g., retina) by the intravitreal injections of solutions, suspensions, biodegradable implants, nanoparticles, or hydrogels. In addition, there are delivery devices that are surgically implanted and tethered to the sclera. Systemic delivery is sometimes used for sight-threatening inflammation, but the GCs must transit the natural blood–aqueous barrier and blood–retinal barrier (which is sometimes compromised during ocular inflammation). Figure created using BioRender.

**Table 1 cells-12-02452-t001:** Ocular conditions treated with glucocorticoids.

Post-operative inflammation
Macular edema (diabetic retinopathy and AMD)
Allergic and hypersensitivity reactions
Uveitis (uveal tract inflammation)
Retinitis
Scleritis and episcleritis
Giant-cell arteritis
Trauma
Herpes zoster ophthalmicus

Note: GCs unmatched for anti-inflammatory/immunosuppressive activities.

**Table 2 cells-12-02452-t002:** Similarities Between GIG and POAG.

Elevated IOP due to increased aqueous outflow resistance at TM
Elevated IOP damages unmyelinated axons at ONH
Morphological changes and remodeling of ONH
Progressive loss of RGC axons and soma
Similar pattern of visual field loss
Associated with morphological and molecular changes to the TM Increased deposition of ECM in trabecular beams and JCT Increased TM tissue stiffness Increased ECM synthesis & crosslinking and decreased degradation Reorganization of actin cytoskeleton to form CLANs Decreased TM cellularity

GIG = glucocorticoid-induced glaucoma; POAG = primary open-angle glaucoma; IOP = intraocular pressure; TM = trabecular meshwork; ONH = optic nerve head; RGC = retinal ganglion cell; ECM = extracellular matrix; JCT = juxtacanalicular tissue; CLANs = cross-linked actin networks.

**Table 3 cells-12-02452-t003:** List of ophthalmic glucocorticoids (steroids).

Generic Name	Drug Name	Concentration	Dosing
Prednisolone-21- acetate	Prednisol Pred Mild Pred Forte	Suspension (0.12%, 0.125%, 1%)	Topical ocular
Prednisolone sodium phosphate		Solution (0.125% and 1%)	Topical ocular
Dexamethasone	Maxidex	Suspension (1 mg/mL)	Topical ocular
	Dexycu	Suspension (517 μg/5 μL)	Posterior chamber injection
	Ozurdex	0.7 mg/implant	Ivt implant
	Dextenza	0.4 mg/insert	Tear duct insert
Dexamethasone sodium phosphate	Ocu-Dex	Solution (1 mg/mL)	Topical ocular
Loteprednol etabonate	Lotemax (0.5%) Lotemax SM 0.38% gel) Eysuvis (0.25%) Inveltys (1%) Alrex (0.2%)	Eye suspension (5 mg/mL) Eye gel (3.8 and 5 mg/g) Eye ointment (5 mg/g)	Topical ocular
Fluoromethalone	FML	Suspension (0.1%)	Topical ocular
	Fluor-Op		
	FML Forte	Suspension (0.25%)	Topical ocular
	FML S.O.P.	Ointment (0.1%)	Topical ocular
Fluoromethalone acetate	Flarex	Suspension (1 mg/mL)	Topical ocular
	Yutiq	Implant (0.18 mg)	Ivt implant
Rimexolone	Vexol	Suspension (1%)	Topical ocular
Difluprednate	Durezol	Ophthalmic emulsion (0.5 mg/mL)	Topical ocular
Fluocinolone acetonide	Retisert	0.59 g/implant	Ivt implant
	Illuvien	0.19 mg/implant	Ivt implant
	Yutiq	0.18 mg/implant	Ivt implant
Triamcinolone acetonide	Xipere	40 mg/mL	Superchoroidal injectable suspension
	Trivaris	80 mg/mL	Injectable suspension
	Triesence	40 mg/mL	Injectable suspension

**Table 4 cells-12-02452-t004:** Topical ophthalmic glucocorticoids: intrinsic potency and IOP response.

Glucocorticoid	IC_50_ (nM)	* Potency	IOP Rise (mmHg)
Loteprednol	1.9	105.8	4.1
Rimexolone	6.2	32.4	6.2
Dexamethasone	8.4	23.8	22.0
Fluoromethalone	9.7	20.8	6.1
Prednisolone	87	2.3	10.0
Medrysone	117	1.7	1.0
Cortisol	201	1.0	3.2

* = potency relative to cortisol, Data from [11,33,34,35].

**Table 5 cells-12-02452-t005:** Animal models of GC-OHT and GIG.

Species	Glucocorticoid	Route
Human-perfused anterior segments	Dexamethasone	In perfusion medium
Non-human primate	Dexamethasone	Topical ocular
Cow—in vivo	Prednisolone	Topical ocular
Cow—ex vivo	Dexamethasone	In perfusion medium
Sheep	Prednisolone	Topical ocular
Rabbits	Dexamethasone Betamethasone Prednisolone-Ac Cortisone Triamcinolone Fluorometholone Loteprednol-etabonate Rimexolone	Topical ocular Subconj injection Topical ocular Subconj injection Subconj injection Topical ocular Topical ocular Topical ocular
Cats	Dexamethasone Prednisolone-Ac Fluorometholone Loteprednol-Etabonate Rimexolone	Topical ocular
Rats	Dexamethasone Betamethasone	Topical ocular
Mice	Dexamethasone	Systemic
	Dexamethasone Acetate	Periocular injection

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
