# Peer review of "Role of Glucocorticoids and Glucocorticoid Receptors in Glaucoma Pathogenesis"

_cells, 2023, doi:10.3390/cells12202452_

Round 1

Reviewer 1 Report

Ref: Role of Glucocorticoids and Glucocorticoid Receptors in Glaucoma Pathogenesis

This review manuscript addresses the function of GCs and GC receptors in GC-OHT and POAG, as well as new research findings and potential therapeutic approaches. The manuscript is well written, organized (table and figures), interesting and insightful; the work has the potential to be accepted after minor corrections. 

However, I have a few minor suggestions for improvement:

1.    In the introduction, the authors provide further detail on the characteristics and function of the trabecular meshwork and the distinction between primary open-angle glaucoma and steroid-induced glaucoma. The importance of the later stated molecular effects of glucocorticoids on trabecular meshwork cells will be clearer to readers given this context.

2.    Although the authors provide a basic history of prior work in the introduction and abstract, they do not specify the review's aim. As a result, I am unsure of what additional information the review provides.

3.    Correct the typing error that reads "3" rather than "4" after the number 3 in the subheadings.

Reviewer 2 Report

This is a well written review summarizing a large body of data about glucocorticoids, their receptors, and relationship to glaucoma.

I don’t have very many recommendations for improving the manuscript; the main one is just that the authors consider how many abbreviations they wish to use and how this may affect readability. There are a lot of abbreviations in the paper that, as a nonspecialist, became distracting. The journal may have rules about how many times a word needs to appear to be abbreviated, but to many readers, I think using POAG, TM and IOP are expected, but PACG, GIG, and others are unfamiliar—this is not a big problem but I think you want your readers to stay with you rather than go have to look up another abbreviation.

Other minor issues-

There are a couple of spelling or grammar that the journal editor should maybe find (line 84 cytoskeletal to cytoskeleton; line 123 prolong to prolonged)

The font for the “Pressure” inside of the eyes in Figure 1 is very small and may not be legible in final form

For Table 1 do the authors wish the first line to be a “Note”?

Overall it’s a well written and helpful contribution to the field.

A couple minor grammar things discussed above
